# Assessing the Travel Health Knowledge of Australian Pharmacists

**DOI:** 10.3390/pharmacy8020094

**Published:** 2020-05-31

**Authors:** Ian M. Heslop, Richard Speare, Michelle Bellingan, Beverley D. Glass

**Affiliations:** 1Pharmacy, College of Medicine and Dentistry, James Cook University, Townsville 4811, Australia; m.bellingan@cqu.edu.au (M.B.); beverley.glass@jcu.edu.au (B.D.G.); 2School of Pharmacy, College of Science, University of Lincoln, Lincoln LN6 7TS, UK; 3Public Health and Tropical Medicine, College of Public Health, Medical and Veterinary Sciences, James Cook University, Townsville 4811, Australia; richard.speare@jcu.edu.au

**Keywords:** pharmacist, travel health, Australia, vaccination, malaria

## Abstract

Worldwide, the numbers of travellers are increasing, with pharmacists having the potential to play a significant role in the provision of pre-travel health services to a large number of these travellers. However, studies examining whether pharmacists have the travel health knowledge to provide these services are limited. This study thus aimed to explore the travel health knowledge of Australian pharmacists. Surveys assessing pharmacists’ knowledge of travel health were distributed through the Pharmaceutical Society of Australia and community pharmacies for self-completion. Overall, the travel health knowledge of participants was found to be good. However, although the majority of participants were aware of the common causes of morbidity and mortality in travel health, some slightly overestimated the prevalence of malaria and were less knowledgeable about the global distribution of some diseases. Most participants also demonstrated an ability to give appropriate advice on the management of traveller’s diarhoea, the selection of appropriate items for inclusion in travel first aid kits, vaccinations, and malarial chemoprophylaxis for travellers visiting endemic areas. This study highlights that Australian pharmacists have the knowledge to deliver travel health advice, with the potential to improve both access and outcomes for travellers.

## 1. Introduction

Many travellers do not seek pre-travel health advice before travelling overseas, and those who do mainly obtain advice from general practitioners (GP) or travel health clinics [1,2,3,4,5]. Reasons given for failing to obtain pre-travel health advice include the travellers’ lack of perception of the risks associated with their destination or difficulties in making appointments with travel health providers [6,7,8,9]. Pharmacists are often seen as being accessible and convenient due to their extensive opening hours, thus offering an attractive alternative to increase the frequency and quality of pre-travel advice sought by some travellers [6,7,8,9]. Pharmacists do play a significant role in the provision of travel health services, although the complexity of the travel health services offered by pharmacists varies somewhat from country to country and region to region [9,10]. In Australia, although most pharmacists will at least respond to simple travel-related requests when asked, most pharmacists play a limited role, and although pharmacist immunisation with a limited range of vaccines is now common, pharmacists are unable to administer travel-related vaccinations such as cholera and yellow fever [10]. However, although pharmacists are generally considered to have good knowledge on commonly encountered areas in travel health, such as travellers’ diarrhoea [11], it has also been suggested that generalist pharmacists may have some knowledge deficiencies, potentially affecting the quality of the travel health information delivered [12,13].

Kodkani et al. [12] examined the knowledge of travel-related health issues amongst Swiss pharmacists and found their overall knowledge to be satisfactory. However, there were areas, such as sun protection, travel vaccinations and malaria prophylaxis, where the information given by the pharmacists was considered lacking. In addition, few pharmacists used specialised travel health information resources when educating travellers and wanted a single, up-to-date, easy-to-use, travel health information resource specifically for use in pharmacies [12]. Kodkani et al. evaluated the pharmacists’ knowledge of the vaccines and antimalarial agents recommended for two common tropical holiday destinations for Swiss tourists (Kenya and Thailand). They found that only 19% and 31% gave accurate advice for malaria protection, and that only 13% and 3% provided appropriate recommendations for vaccinations for Thailand and Kenya, respectively [12]. However, 50% of pharmacists said that in practice they would have consulted standard reference materials before answering such questions, which was confirmed on follow-up, where 74% and 93% of pharmacists gave correct advice for malaria protection for Thailand and Kenya respectively [12].

Another study examining the quality of travel advice given to international travellers by pharmacists was performed in Portugal [13] where most of the responding Portuguese pharmacists (93%) did not have any additional training in travel health. The study found gaps in pharmacists’ knowledge of travel health and inaccuracies were found in the advice given by the pharmacists [13]. The researchers concluded that the pharmacists in the study required more training in the area of travel health as the advice they gave was often incomplete and/or incorrect [13]. Most of the responding pharmacists said that they would like more training or information to use in their practice [13]. Similar to the Swiss study, it was recommended that pharmacists have greater access to specialised training in travel health [13].

More recently, Bascom et al. [14] examined the baseline travel health knowledge of pharmacists’ in Alberta, Canada and their confidence to provide advice, because confidence is required for the successful integration of knowledge in practice. In addition, they also investigated pharmacists’ preferred means of obtaining education on travel health. Although the mean knowledge score was only 27%, two thirds of the pharmacists felt confident that they would be able to source the correct information. However, only 21% were confident in their overall ability to provide travel health advice highlighting the need for both undergraduate and continuing education training programs to accommodate this expanding area of practice [14]. The aim of this study was to examine the current level of knowledge in travel health of Australian Pharmacists in order to inform training requirements.

## 2. Materials and Methods

### 2.1. Study Design and Participants

A cross-sectional survey of Australian pharmacists was performed with a self-completion questionnaire formatted into an electronic e-survey and a postal survey using SurveyMonkey^®^ and Microsoft Word^®^ respectively. An invitation to participate in the e-survey was posted to all members of the Pharmaceutical Society of Australia (PSA) in a PSA weekly newsletter. The invitation contained a hyperlink to the electronic e-survey and a participant information leaflet. The postal survey was distributed to a sample of 601 Australian community pharmacies. To ensure that more populous states and territories were not over-represented in the sample, pharmacies were selected using a systematic random sampling technique by selecting every thirteenth pharmacy from the pharmacy businesses in the alphabetical listing of the current Yellow Pages^®^ Business Directory for Australia. The e-survey and the postal survey were each open for a 6-week period. In an attempt to improve response rates, participants were given the opportunity to enter an anonymous prize draw to win one of two Apple iPods^®^.

### 2.2. Questionnaire Design and Testing

The questionnaire contained a combination of multiple-choice questions (MCQs), multiple answer questions (MAQs), and small vignettes. It was designed to collect respondents’ demographic data and to test their knowledge of selected travel-related health issues. Topics included the common causes of morbidity and mortality in travellers, the causes and management of Traveller’s diarrhea, and the selection of appropriate vaccinations, malaria prophylaxis, and first aid items for travellers. Participants were asked to complete the questionnaire without further research or reference to information resources. To ensure validity and reliability, to reduce bias, and to allow comparison with other studies, some of the questions used in the self-completion questionnaire were based on similar questions used in other surveys [12,13]. In addition, before the questionnaire was distributed, it was pre-tested by a group of 5 pharmacists for understanding, readability, and to ensure timely completion. Only minor grammatical changes were made prior to distribution.

### 2.3. Data Analysis

Survey responses were collated in Microsoft^®^ Excel^®^ spreadsheets and the IBM^®^ SPSS Statistics Package^®^ (Version 22) was used for statistical analyses.

### 2.4. Ethical Considerations

An ethical review was performed by James Cook University Human Research and Ethics Committee and approval was granted (Approval No: H3182). In addition, approval to send a postal survey to Australian community pharmacies was granted by the Survey Approval Program of the Pharmacy Guild of Australia (Approval No: 755).

## 3. Results

### 3.1. Respondent Characteristics

A total of 208 pharmacists completed the knowledge assessment. A definitive response rate for the electronic survey could not be calculated as it was unclear how many PSA members read the newsletter. However, the response rate for the postal survey was 14.3%. Participants were predominantly female (67%), below the age of 50 years (75%), registered for less than 30 years (81%) and resided in regional metropolitan areas or in state capitals (76%). The majority held full-time positions (71%) predominantly in community pharmacy (77%). In contrast, others worked in hospital pharmacy (8%), military pharmacy (4%), academia (1%), or in miscellaneous areas (10%). Most had standard entry level pharmacy qualifications (82%), and although some had postgraduate qualifications (11% with postgraduate certificates, 7% with Master degrees and 1% with doctorates), the vast majority (97%) had no formal training in travel health. Finally, 69% of participants stated that they provided travel-related health advice or services. Although for the majority, their travel health workload was low with 70% of participants advising less than 2 travellers per week and 83% spending less than one hour per week providing travel health advice.

### 3.2. Causes of Morbidity and Mortality and Disease Distribution

Participants were asked to select the commonest cause of mortality for travellers to low and middle income countries from a range of conditions. Over half (56%) identified motor vehicle accidents to be the most common causes of mortality, followed by malaria (19%) and myocardial infarction (12%). Fewer participants selected cholera (6%), typhoid (4%) and HIV (2%). In addition, participants were requested to select the four most common travel-related health problems experienced by Australians travelling to low and middle income countries and the 10 most commonly selected health problems are presented in Table 1.

Finally, participants were given a list of 5 countries (Brazil, Kenya, India, Thailand, and Japan) and asked to identify whether yellow fever, malaria, and typhoid were prevalent in each country. The results are summarised in Table 2.

### 3.3. Traveller’s Diarrhoea

Participants were asked to select the most common causative organism for traveller’s diarrhoea. Over half (52%) of participants selected Enterotoxigenic *E.coli* (ETEC) followed by *Giardia intestinalis* (19%), *Salmonella* spp (17%), rotaviruses (12%), and *Campylobacter pylori* (1%). Vignette 1 then presented participants with a traveller’s diarrhoea scenario and asked them to select their preferred course of action. The vignette and participants’ preferred treatment options are summarised in Table 3.

### 3.4. Items for Traveller’s First Aid Kits

From a list of standard first aid items, participants were asked to select five items that they would recommend to the traveller in vignette 1 to take with him on his journey. The fifteen most commonly selected items are summarised in Table 4.

### 3.5. Vaccines

Vignette 2 asked participants to select the vaccinations they would recommend for a 30 year old woman planning a 4 week safari in Kenya. Participants were presented with a list containing the Centers for Disease Control and Prevention (CDC) recommended vaccinations for the destination plus a number of other common travel vaccines. The majority of participants selected vaccines that are included in the CDC vaccination recommendations for Kenya [15] and that are also commonly included in the vaccination recommendations for other low and middle income countries. These include vaccines such as typhoid (89%), hepatititis A (88%), hepatitis B (88%), tetanus (79%), yellow fever (75%), and cholera (62%). However, fewer participants recommended vaccinations such as polio (49%), rabies (35%), and meningococcal disease (28%).

Vignette 3 asked participants about post-exposure prophylaxis for suspected rabies exposure in a 35-year old woman visiting a Cambodian temple complex. Participants were asked to select their preferred course of action and the results are summarised in Table 5.

### 3.6. Malaria

Participants were presented with five statements relating to malaria and asked to select the incorrect statement. The participants selections are summarised in Table 6:

The final vignette (Vignette 4) presented participants with the scenario of a young female teacher travelling to Thailand to work in a rural location. The scenario contained a number of complicating factors and participants were asked to give a recommendation on the most appropriate agent for malarial chemoprophylaxis. The largest number of participants (43%) recommended doxycycline followed (in order by) atovaquone/proguanil (34%), mefloquine (17%), chloroquine (3%), and artemether/lumefantrine (3%).

## 4. Discussion

The number of studies examining the travel health knowledge of pharmacists is relatively low [12,13,14] and none appear to have examined the travel health knowledge of pharmacists in Australia. This study examined the travel health knowledge of a sample of Australian pharmacists and, as with comparable studies, some knowledge gaps were identified. A total of 208 Australian pharmacists participated in our study and, although low, the number of participants was comparable to the number of participants (84-251 pharmacists) in similar studies [12,13,14]. Kodkani et al. gave little detail about the demographic characteristics of the participants in the Swiss study [12], however there are similarities between the demographic characteristics of the participants in this study to those of other comparable studies [13,14]. That said, the proportion of female participants was slightly greater in the Portuguese study by Teodosio et al. (79.9%) [13]. Likewise, the travel health workload of most participants in this study was low, which was comparable to the other studies, with the majority of participants in the Swiss and Portuguese studies (56% and 87.6% respectively)) only giving travel health advice to up to three travellers per month [12,13]. Similarly, Bascom et al. reported that 76% of participants in the Canadian study counselled up to one traveller per month [14]. Finally, Bascom et al. also reported that 43% of participants had not received any formal travel health training [14]. However, in this study, this was doubled with 97% of participants not having received formal travel health training, which was comparable to the findings (93.2%) of the Portuguese study [13]. However, the Portuguese and Canadian studies reported that while many participants had not received formal training, many did attempt to stay informed and up to date using a variety of information, self-study and/or online resources. Unfortunately, the participants in this study were not asked whether they undertook self-study or continuing professional development to maintain currency [13,14].

Background knowledge of common causes of morbidity and mortality amongst travellers was investigated in this study. Epidemiologically, the most common causes of mortality among visitors to low- and middle-income countries are accidents/trauma or cardiovascular disease with mortality due to infectious diseases being relatively rare [16,17]. Malaria is the most prevalent infectious cause of mortality amongst travellers [16,17]. Likewise, the most common causes of morbidity in travellers are traveller’s diarrhoea and influenza, with other infectious and tropical diseases having a much lower prevalence. Just over two-thirds of participants (67.8%) in this study were aware that accidents and cardiovascular disease are the most common causes of mortality in travellers. The participants ranking of common health problems experienced by travellers (Table 1) is similar to their actual prevalence [16,17] However, participants ranked some infectious diseases, such as cholera slightly more highly than their actual prevalence. In addition, participants also ranked jet lag and motion sickness relatively highly, which could be because pharmacists tend to give advice about these conditions more frequently. Finally, participants were asked to identity countries in which malaria, yellow fever and typhoid were prevalent from a selection of five destinations. Table 2 shows that the vast majority of participants (greater than 93%) were aware that these diseases were not prevalent in Japan. Likewise, for Kenya, Thailand, and India, most participants (70.7%–87.5%) selected correct options. However, only 45%–54% of participants selected correct options for Brazil, possibly because participants are less familiar with this destination as relatively fewer Australians visit Brazil [18]. Finally, participants appeared relatively less knowledgeable about the global distribution of yellow fever compared to malaria or typhoid as over a third (35%) and almost a fifth (18%) of participants incorrectly thought that yellow fever was prevalent in India and Thailand respectively. Likewise, Teodosio et al. examined pharmacists’ knowledge of yellow fever vaccination requirements for visitors to Portuguese-speaking countries. They found that 26.8% of participants reported that they were unfamiliar with the topic and only 8.8% could correctly indicate whether yellow fever was or was not a risk in Portuguese-speaking countries [13].

Traveller’s diarrhoea (TD) is one of the most common travel-related health conditions. Kodkani et al. [12] found in their telephone interviews that only 59% of the Swiss participants spontaneously recommended rehydration therapy for TD whereas 100% recommended the use of antimotility agents and only 34% recommended the use of antibiotics. These figures however changed to 90%, 96%, and 39%, respectively, in their follow-up written survey. In comparison, Teodosio et al. [13] found that Portuguese participants were more likely to recommend the use of antibiotics (57%) than antimotility agents (53%) and only 56% of Portuguese participants recommended the use of rehydration therapy. In our study, all of the treatment options presented to participants included rehydration therapy. However, the two most commonly selected treatment options were rehydration therapy alone (31% Option 1) or a combination of rehydration therapy, antibiotic and antimotility agent (30% Option 3), both of which are clinically justifiable [19,20]. In total, 56% of participants chose treatment options that involved the use of antibiotics (Options 3, 4, and 5) and 46% chose treatment options that involved the use of antimotility agents (Options 2 and 3). This implies that the participants in this study are more likely to recommend the use of antibiotics in the management of TD than those in the Kodkani study and equally as likely as the participants in the Teodosio study. However, most participants who recommended the use of antibiotics recommended the use of norfloxacin, an appropriate antibiotic [19,20], and only 3% of participants recommended the use of doxycycline, an inappropriate antibiotic [19,20], suggesting that participants knew how to appropriately manage TD with rehydration therapy, antibiotics and/or antimotility agents [19,20].

The provision of advice relating to the first aid items and OTC remedies travellers should carry is recognised as a key role for travel health pharmacists [20]. When listed in order of importance, oral rehydration salts and antidiarrhoeal medications were rated relatively highly. Whereas, in contrast, simple analgesics, which are often mentioned as being the most useful items for travellers to carry [21,22,23] were only rated sixth in order of importance in this study. That said, the 10 most frequently selected items chosen by the participants are included in the recommendations for simple travel first aid kits, thereby demonstrating that respondents are capable of advising travellers on the most appropriate items to carry [21,22,23].

Participants’ knowledge about travel vaccines required in relation to travel to Kenya for a 4 week safari were compared with the current CDC and Medical Advisory Service for Travellers Abroad (MASTA) vaccination recommendations for that destination [15,24]. At the time of the study these were; typhoid, hepatitis A and B, yellow fever, polio, rabies and meningococcal meningitis plus the standard childhood vaccinations. These recommendations were somewhat aligned with those recommended by the majority of the participants (between 75% and 89%), whose advice included tetanus, typhoid, hepatitis A and B, and yellow fever, with 62% of participants also stating that they would recommend cholera vaccine. Fewer participants recommended vaccinations for polio (49%), rabies (35%) and meningococcal disease (28%). Importantly, although participants were asked to answer questions without referring to reference material, it is unclear how many did not do so. Kodkani et al. also asked respondents in their surveys about the vaccination requirements for travellers to two destinations (Thailand and Kenya) [12]. Similarly, they found that many respondents wanted to consult information resources before answering, but that in both the telephone and written survey that many did not give correct advice [12]. With regard to Kenya, only 3% of participants in Kodkani et al.’s study gave accurate advice, whereas 62% said they would firstly consult information resources and 8% gave inaccurate advice. In the written survey, the number giving accurate advice rose to 43%. However, the number giving inaccurate advice also rose to 47% [12].

Management of a dog bite and dealing with the risk of rabies resulted in participants (56%) selecting the management option recommended in the Australian Immunisation Handbook [25]. Only 4.3% of the participants chose options in which they would not refer the affected person for further treatment.

There was some concern that 27% of participants appeared to believe that vitamin B_1_ (Thiamine) is effective in decreasing mosquito bites, when evidence is limited [26]. However, in relation to the selection of appropriate agents for malaria chemoprophylaxis, the majority (77%) selected clinically justifiable options [27]. In the scenario presented, 23% of participants chose less than ideal options, because artemether/lumefantrine is used mainly in the treatment of malaria [27], and there is significant resistance to the agents chloroquine and mefloquine in the area being visited [27]. However, 77% chose either atovaquone/proguanil or doxycycline. Atovaquone/proguanil, would be the clinically preferred option as it will not interfere with any of the traveller’s current medications or co-morbidities [27]. However, it is also relatively expensive, which is possibly why it was only selected by a third of participants (34%). Doxycycline, which may be slightly less ideal than atovaquone/proguanil, is also a lot cheaper which is probably why it was selected by a slightly greater number of participants (43%) [27]. However, the selection of either atovaquone/proguanil of doxycycline could be clinically justified [27]. Kodkani et al. [12] also found the knowledge of Swiss pharmacists in this area to be satisfactory with over 95% being able to name the most important bite prevention methods. However, they too noted that up to 20% of pharmacists also recommended thiamine for the prevention of mosquito bites. When making recommendations for chemoprophylaxis, only 27% and 35% of all respondents were willing to give immediate advice on appropriate chemoprophylaxis for Thailand and Kenya, respectively, in the telephone survey and 19% and 31% of all participants gave acceptable answers [12]. However, in the follow-up written survey, Kodkani et al. [12] report that this increased and 74% (Thailand) and 93% (Kenya) of all respondents gave acceptable answers as they could refer to information resources.

This study has some limitations. Firstly, the survey was distributed by two methods and, although pharmacists were asked not to do so, a pharmacist could have completed both surveys. However, there was no evidence to suggest that this occurred. Secondly, participants, may represent those pharmacists that have an interest in travel health, making it difficult to generalise the results to all Australian pharmacists. However, this is a limitation that may be true of many surveys

## 5. Conclusions

This study is one of the first to examine the travel health knowledge of a sample of Australian pharmacists. The majority were aware of the common causes of morbidity and mortality in travel health, and of the health risks associated with some common destinations for Australian travellers. However, they appeared to be slightly less knowledgeable about the prevalence of malaria, the global distribution of yellow fever, and some other travel-related infectious diseases. Their knowledge of traveller’s diarrhoea, its management, and appropriate items for inclusion into travel first aid kits was comparable to the findings of studies in other countries.

From an Australian public health perspective, the pharmacists in this study importantly demonstrated an awareness of the vaccination requirements for common destinations for Australian travellers and a capability to select appropriate malarial chemoprophylaxis for visitors to endemic areas. They also demonstrated an ability to judiciously and appropriately recommend the use of antibiotics in the management of traveller’s diarrhoea. This adds credence to the case to allow the supply of a wider range of vaccines, antimalarials, and a limited range of antibiotics for use in travel health from Australian pharmacies.

Future research using validated questions, such as those developed by the International Society of Travel Medicine for their Certificate of Travel Health (CTH), would be beneficial to compare the travel health knowledge of Australian pharmacists with those of other countries. Findings could be used to develop training resources, accreditation tools, and to further develop the extended role of pharmacists.

## Figures and Tables

**Table 1 pharmacy-08-00094-t001:** Pharmacists’ perceptions of the most common causes of morbidity in Australian travellers.

Health Problem	Number (%) of Participants (n = 208)
Diarrhoea	192 (94%)
Jet Lag	137 (66%)
Acute Respiratory Infection	118 (57%)
Motion Sickness	105 (51%)
Malaria	73 (35%)
Hepatitis A	61 (29%)
Cholera	43 (21%)
Typhoid	40 (19%)
Hepatitis B	27 (13%)
Gonorrhoea	15 (7%)

**Table 2 pharmacy-08-00094-t002:** Pharmacists’ perception of disease prevalence in 5 countries.

	Number (%) of participants identifying the disease as prevalent in the country		
Disease	Yellow Fever	Malaria	Typhoid
Brazil	112 (54%)	109 (52%)	94 (45%)
Kenya	158 (76%)	154 (74%)	167 (80%)
India	72 (35%)	154 (74%)	176 (85%)
Thailand	38 (18%)	182 (88%)	147 (71%)
Japan	14 (7%)	2 (2%)	13 (6%)

Note: Correct responses shaded.

**Table 3 pharmacy-08-00094-t003:** Pharmacists preferred treatment options for traveller’s diarrhoea.

**Vignette 1: Jeff is a 26 year old mountaineer trekking in the Himalayas for 1 month. He develops diarrhoea and has had four loose bowel motions in the last 24 h with nausea, abdominal cramps and faecal urgency. His doctor has supplied some medications including Gastrolyte^®^ tablets, Loperamide 2 mg capsules, Norfloxacin 400 mg tablets and Doxycycline 100 mg tablets. Which of the following statements best summarises how you would recommend he treats his diarrhoea? (Select one answer)**	
**Treatment Option**	**Number (%) of Participants (n = 208)**
1. No active treatment, just maintain hydration using Gastrolyte^®^ and allow the diarrhoea to take its course.	64 (31%)
2. Start treatment immediately with loperamide whilst drinking plenty of fluids.	27 (13%)
3. Start treatment immediately with one dose of norfloxacin 800 mg plus loperamide whilst drinking plenty of fluids.	63 (30%)
4. Start treatment immediately with one dose of norfloxacin 800 mg whilst drinking plenty of fluids.	48 (23%)
5. Start treatment immediately with one dose of doxycycline 200 mg plus loperamide whilst drinking plenty of fluids.	6 (3%)

**Table 4 pharmacy-08-00094-t004:** Top 15 first aid items recommended by participants.

First Aid Item	Number (%) of Participants (n = 208)
Oral rehydration sachets/tablets	166 (80%)
Small range of bandages, dressings and tapes	127 (61%)
Antidiarrhoeal agent	114 (55%)
Sunscreen	107 (51%)
Insect repellent	92 (44%)
Paracetamol or aspirin tablets	89 (43%)
Iodine solution	73 (35%)
Appropriate antimalarials	57 (27%)
Norfloxacin tablets	48 (23%)
Broad spectrum antibiotic	46 (22%)
Metoclopramide tablets	30 (14%)
Antiseptic cream	29 (14%)
Sharps kit	17 (8%)
Salbutamol inhaler	16 (8%)
Antihistamine tablets	14 (7%)

**Table 5 pharmacy-08-00094-t005:** Pharmacists preferred treatment options for a dog bite.

**Vignette 3: Jayne is a 35-year old woman bitten by a dog whilst visiting a temple in Cambodia. The dog appears to be behaving normally but the dogs teeth punctured her skin. What would be the best course of action for Jayne to take? (Select one answer)**	
**Treatment Option**	**Number (%) of Participants (n = 208)**
1. Vigorously wash the area with soap and water and then apply povidone iodine and seek urgent medical attention to get the urgent administration of rabies vaccine.	46 (22%)
2. Vigorously wash the area with soap and water and then apply povidone iodine and seek urgent medical attention to get the urgent administration of rabies vaccine and rabies immunoglobulin.	116 (56%)
3. Vigorously wash the area with soap and water and then apply 70% alcohol and povidone iodine and seek urgent medical attention to get the urgent administration of rabies vaccine.	37 (18%)
4. Wash the area with soap and water only.	9 (4%)
5. No action is required.	0 (0%)

**Table 6 pharmacy-08-00094-t006:** Pharmacists understanding of malaria prevention.

**In relation to the prevention of malaria, participants were requested to select the incorrect statement:**	
**Statement**	**Number (%) of Participants (n = 208)**
1. Oral vitamin B1 is not effective in decreasing the number of mosquito bites.	56 (27%)
2. The ideal mosquito repellent should contain 20–30% DEET (Diethyltoluamide).	19 (9%)
3. When needing to apply sunscreen and mosquito repellent together it is better to apply the repellent first and wait 20 min before applying the sunscreen.	122 (59%)
4. Travellers should wear covered shoes and loose fitting, long trousers and long-sleeved, light coloured clothing between dusk and dawn in malarial areas.	8 (4%)
5. Ideally travellers should sleep in air conditioned or well screened rooms or under treated mosquito nets.	3 (1%)

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
