# Peer review of "Assessing the Travel Health Knowledge of Australian Pharmacists"

_pharmacy, 2020, doi:10.3390/pharmacy8020094_

Round 1

Reviewer 1 Report

Overall, this is an interesting and timely article, and if the authors are correct, would be the first of its kind for Australian Pharmacists which would greatly add to the literature However, I do have some significant concerns that the authors must address before I feel its suitable to publish (in no particular order):

1.  I would like to see an analysis of pharmacist knowledge by demographic/background characteristic to see if anything is statistically significant and why. 

2. A discussion of the current state of Australian pharmacy practice and Australian pharmacy education would be especially helpful in the introduction section for those not familiar with this Country.

3. Is this knowledge assessment generalizable to ALL Australian pharmacists? Why or why not? In other words, does this assessment definitively define knowledge so that if someone scores high on it, their knowledge would not be questioned or vice versa for someone scoring low?

4. One of the biggest issues I have with this paper is the possibility for survey contamination since both an electronic and paper survey were used. What did the authors do to minimize this or remove duplicate answers? If nothing was done, then the results are quite questionable and would not be sure how valid they actually are. In this case, I would suggest that the authors conduct a post hoc review to identify duplicates

5. I understand that the assessment was validated on face and content, but why wasnt something like ISTM's CTH practice questions used instead? This would seem to use a more reliable and validated source for questions. 

6. What does it mean when the authors say that the overall knowledge of pharmacists was "good"? What does good mean? How is it determined? 

7. The conclusion section needs to be reworked. Presently, it reads more like something you would find in the discussion section. Conclusion should focus on the bigger picture or next steps that result from this work. 

Minor issues:

  1. Lines 29-30: please provide stats to back up this statement
  2. Line 32: citation needed at end of sentence 
  3. Line 40: what does satisfactory mean?
  4. Lines 42-44 (starting with "in addition..."): please clarify/reword. This is confusing as is. 
  5. I dont think that you need to continuously cite the same reference over again in the same paragraph. See lines 44-51 for an example. 
  6. Line 60: I think you mean citation #12
  7. Lines 76 to 80: was this postal survey done in addition to the hard copy mailed out to the PSA? Im a bit confused by who was sent what survey and think you need to clarify this. Also, please give additional details about the "systematic random sampling technique" used. 
  8. How many people were sent the survey? What is your response rate to it? Did it differ between emailed and hard copy? If so, why?
  9. Clarify whether an incentive was used for survey completion or not. 
  10. Respondent characteristics: what is standard entry level mean? 
  11. The first line or so in the results section that describe the assessment, should actually be placed in methods. Only results should be reviewed in the results section. 
  12. Line 145: Im not sure you need to cite ref #14 if this is common knowledge. 
  13. Table 5: whether the participant can identify an incorrect answer does not necessarily mean that they can identify the correct answer. Usually, questions that ask to identify the incorrect answer preform more poorly in post hoc reviews. 
  14. Lines 166-176: these paragraphs look like they belong in the intro and not the discussion section.  

Reviewer 2 Report

Thank you for your submission! 

Methods: It was a well thought out research project.  In your demographic data did you ask how many years the participants practiced as a pharmacist?  It might give the reader a sense of how much experience the participants had - younger versus a pharmacist with many years experience.  You did state an age but it is clearer if you report how many years of experience they have.

Results: What other area(s) of pharmacy practice answered the survey?  It was stated 77% were community.  Were the rest hospital based or in an ambulatory clinic setting? 

Your references are correctly cited and all were checked. 
